# STARLING: Self-supervised Training of Text-based Reinforcement Learning Agent with Large Language Models

## Abstract

Interactive fiction games have emerged as an important vehicle to improve the generalization capabilities of language-based reinforcement learning (RL) agents. Existing environments for interactive fiction games are domain-specific or time-consuming to generate and do not train the RL agents to master a specific set of skills. In this work, we introduce an interactive environment for self-supervised RL, *STARLING*, for text-based games that bootstraps the text-based RL agents with automatically generated games (based on the seed set of game ideas) to boost the performance and generalization capabilities to reach a goal of the target environment. These games let the agent hone their skills on a predefined set of tasks. We create and test an environment with 100 games, generated using this automated framework that uses large language models (GPT3) and an interactive fiction game engine (based on Inform7) to provide the user with the ability to generate more games under minimal human supervision. Experimental results based on both the human participants and baseline text-based RL agents reveal that current state-of-the-art text-based RL agents cannot use previously learned skills in new situations at the level humans can. These results enforce STARLING's potential to serve as a sandbox environment for further research in self-supervised text-based RL.

## 1 Introduction

Interactive fiction games such as Zork can be utilized as an important test-bed to improve the generalization capabilities of text-based reinforcement learning (TBRL) agents Hausknecht et al. (2020); Jansen (2022). In these games, both the observed state of the game and the actions taken are in natural language. To play these games, the agents (or human players) need to understand the observed text from the environment and take relevant action toward the goal. These games encourage agents to understand the underlying state of the game and take actions to interact with the environment. In order to be successful, agents must use previously learned skills in new situations to complete an overarching goal. Current environments of interactive fiction games suffer from two major problems. First, environments such as TextWorld Commonsense measure simple commonsense reasoning based on one-hop relationships between entities (e.g., apple → refrigerator) Murugesan et al. (2021a) but lack game complexity (besides a fewer number of games) to learn skills and generalize to novel domains. Second, environments such as ScienceWorld (even though many variations of task-based games are available) and Jericho are domain-specific so agents that play these environments may perform well while conducting specific tasks like completing science experiments but lack generalized skills to apply them to other situations Wang et al. (2022); Hausknecht et al. (2020). Most importantly, in order to generate a large number of games to train the RL agents to master skills in these environments, we will have to employ human annotators to manually design, generate, and deploy the game. Therefore, the purpose of this work is to develop an efficient approach that generates a large amount of text-based games to train the RL agents to master the desired skills and excel at the target environments such as TWC, ScienceWorld, etc.

As developing a set of text-based games is a time-intensive manual process, we propose Self-supervised Text-bAsed RL learnING, "STARLING", an interactive environment that utilizes Large Language Models (LLM) and an integrated interactive fiction game engine (Inform7 Nelson (2006))

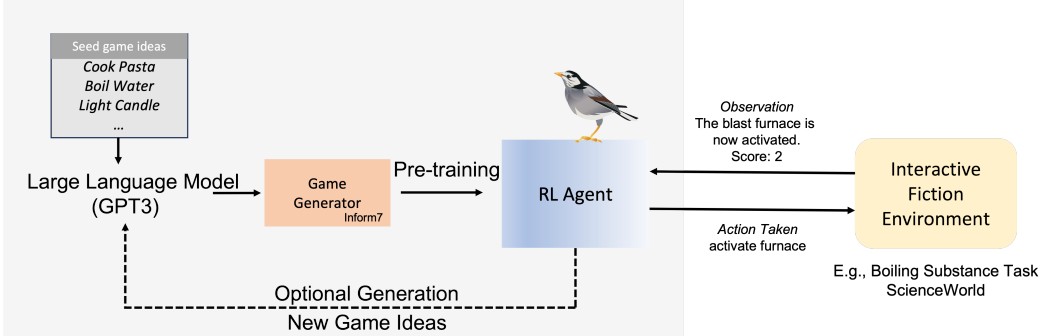

Figure 1: Architecture diagram for Self-supervised Text-based Reinforcement Learning using LLM (STAR-LING).

to easily produce games in any domain. We generate a set of 100 text-based games using GPT3 Brown et al. (2020b) based on the input game ideas (*seed* list) that emphasize the need for the everyday skills such as *boiling water, cooking pasta, etc.* in (pre-)training text-based RL agents. These games require agents to use a specific sequence of actions for achieving the goal and successfully completing the game. For example, while cooking pasta, an agent must first *gather the ingredients*, *fill pot* with water, *boil the water*, and *put the pasta* in the pot. We then deploy the pre-trained RL agent on the target environments. This novel game-generation method can easily be used by others to create their own games and be adapted for future applications to build challenging RL agents in various domains. Figure 1 show the overview of the proposed approach for self-supervised text-based reinforcement learning using LLM.

## 2 SELF-SUPERVISED TEXT-BASED RL

Self-supervised RL involves bootstrapping RL agents with auxiliary tasks in an unsupervised or semi-supervised setting to accelerate learning and generalize in the target tasks. With the recent interest in LLMs, in this paper, we consider LLMs as an alternative option to pre-train an RL agent with minimal human supervision. Unlike in the other text-based environments such as TextWorld Barnes et al., TextWorld Commonsense (TWC) Murugesan et al. (2021a), Jericho Hausknecht et al. (2020), ScienceWorld Wang et al. (2022), we utilize the skill generation capability of the large language models Huang et al. (2022) to automatically generate text-based games based on the input game ideas with minimal human supervision.Our proposed approach for self-supervised TBRL, STARLING is an interactive text-based environment with assistance from LLM and enables the text-based RL agent to hone their extra-curricular skills [1]. In this paper, we assume a seed list of game ideas are already available as an input to STARLING. These game ideas are chosen to exhibit specific skills either for creating a generalized agent or targeting domain-specific environments. Optionally, RL agent can generate a new set of game ideas specific to the target domain to improve it's performance.

### 2.1 CONSTRUCTING PRE-TRAINING GAMES

In this section, we briefly describe how we generate the pre-training games from the game ideas using LLM and Inform7. Given the set of game ideas (seed) to LLM Brown et al. (2020a), we design a method that procedurally generates text-based games based on the interactive fiction game engine. In this paper, we use GPT3 as our LLM. Inform7 is an interactive fiction programming language that allows users to create interactive fiction games using natural language instructions Nelson et al. (2013). Previous text-based environments such as TextWorld, Jericho, ScienceWorld, etc. use Inform7 (in the backend) to generate a handful of text-based games manually that require agents to explore the environment and take a sequence of actions to complete a goal such as *cooking a pasta*. Based on our observation from these environments, we find that the game generation can be modularized into four parts: setup, object creation, custom action, and reward assignment:

---

[1]In this work, we define the skills based on the auxiliary task such as "boil <object>", "fill <container>", "cook <object>", etc.

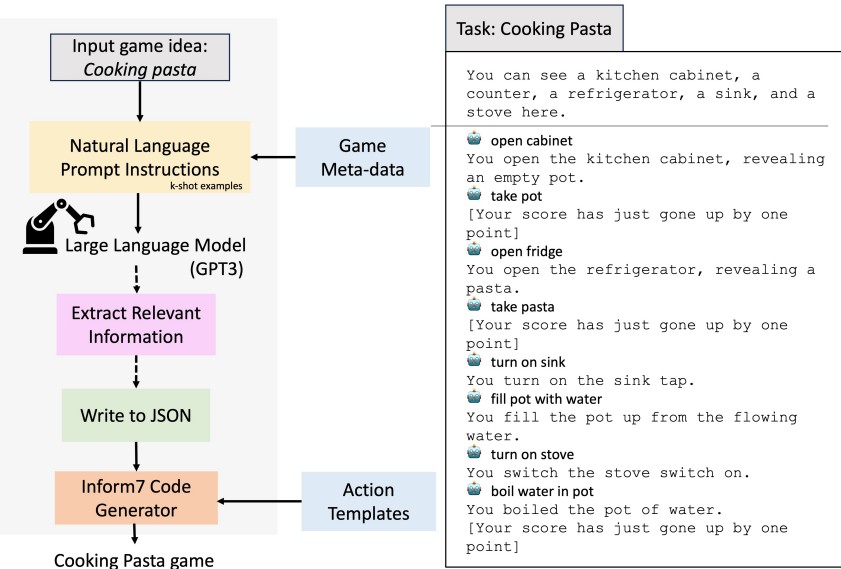

Figure 2: (left) Workflow of the STARLING Game Generator using large language model (GPT3) and game-specific statistics for the 100 pre-training games generated by STARLING. (right) Example text-based RL agent play-through of cooking pasta game.

1. *Setup* - defines basic properties about the game such as the room, entity types, any external libraries (inform7), etc.

2. *Object Creation* - creates in-game entities such as bread or jelly. Each entity is placed in its proper location like the refrigerator or cabinet and assigned properties such as portable, open, or closed.

3. *Custom actions* - defines actions not native to Inform7. Each custom action checks for the pre-conditions and then executes the action by initiating the relevant state changes, and returning the proper observations to the agent. We utilize predefined action templates to incorporate custom actions during the game generation.

4. Rewards - assigns reward value for gathering the necessary entities and completing custom actions to achieve the goal. Once all the rewards are collected for each game, the game ends.

Figure 2 (left) shows the overview of the game generation using STARLING. When we feed a game idea from the seed list, STARLING prepares a prompt using natural language instruction and example game metadata as shown in Figure 8(a) (supplementary), with information about the setup, objects, custom actions, and rewards required for the game idea. We input this prompt to an LLM which generates the requested information as shown in Figure 8(b)(supplementary). We initiate each prompt for a game idea with the necessary objects that the agent needs and agents must collect those objects and use them to cook, clean, build, or complete the high-level task. In order to be successful in these games, agents must understand the properties, location, and affordances of objects in addition to the specific sequence of actions needed to accomplish the task. We write the output from the GPT-3 output into a JSON file as shown in Figure 12 (supplementary). The objects, actions, and tasks from the GPT-3 output correspond to the entities, custom actions, and verbs sections of the JSON file respectively. At this stage, the user may update or change game information in the JSON file. If the user approves the game metadata in the JSON file, we write and compile the Inform7 code based on the JSON file into an Inform7 game for a given game idea. If the user approves the game-related data in the JSON file, we write and compile the Inform7 code based on the JSON file into an Inform7 game for a given game idea. We compile this code using the Glulx [2] interpreter for interactive fiction games. In Figure 2 (right), we show the text-based RL agent play-through of a sample game generated by STARLING.

---

[2] https://en.wikipedia.org/wiki/Glulx

| Game-specific Statistics | |
| --- | --- |
| *Min. # Actions* | $7.36 \pm 2.53$ |
| *Avg. Rewards across games* | $4.08 \pm 1.57$ |
| *Num. Skills per game* | $2 \pm 1$ |

## 2.2 PARSING LLM RESPONSE

Since the response generated by LLM may not strictly follow the desired format, we follow additional steps to mitigate the irrelevant content in the response from LLM. First, we request a specific set of game-related data from LLM in a slot-filling style text generation to reduce the amount of long unstructured text generation Rakotonirina et al. (2022). Since LLM are good at instruction-following when few-shot examples (input-output pairs) of a similar problem are given as a part of the prompt input, we add k-shot examples (k=3) to guide the LLM to generate a response. Figure 8(a) (supplementary) shows one of the three examples given as a part of the input prompt. Finally, during game compilation, the game compiler verifies whether the information extracted from the response adheres to the Inform7 programming language syntax. In addition, the pipeline for game generation provides an option for users to review the generated JSON file before the compilation. When the generated game files still contain irrelevant content, we repeat the text generation multiple times to get the desired response from LLM.

## 2.3 GAME INSIGHTS

Using the above approach repeatedly, we built a set of 100 games with minimal human supervision for training and evaluating the text-based RL agents with skills. In the table above, we show the statistics for generated games such as the average number of actions available per step, the average rewards across the 100 games, and the average number of skills needed to complete each game. These games have multiple sub-tasks which indicate that agents must utilize at least 2 skills (on average) for each game in the correct order. For example, as shown in Figure 2 (right), one skill involves *gathering the ingredients* which must be done before the second skill of *cooking pasta* is executed. The minimum number of actions indicates that agents must take approximately 7 actions on average to complete each game, though some of these actions do not necessarily have to be completed in order (e.g. the agent can "turn on the stove" before "fill the pot with water" and vice versa). Games in TWC only require agents to gather objects and take actions to place them in their commonsense locations. These actions can often be completed in any order, whereas generated games, such as *cooking pasta*, require agents to gather objects and use other related skills in a specific sequence to achieve the final goal.

## 3 EXPERIMENTS

In this section, we report the experimental results of the proposed approach: STARLING. We pre-train the RL agent on the generated 100 games (train split). We evaluate the pre-trained agent (STARLING) on three benchmark environments for text-based games: TextWorld Commonsense (TWC) with easy, medium, and hard difficulty levels Murugesan et al. (2021a) 2) ScienceWorld with 4 tasks and variations Wang et al. (2022). 3) Zork1 from Jericho.

## 3.1 TEXT-BASED RL AGENT

In this section, we briefly describe the text-based agent used for all the experiments. Based on the recent observation that using LLM to learn the underlying representation of text in the environment does not necessarily improve the performance (Wang et al., 2022), we follow previous works He et al. (2016); Murugesan et al. (2021a); Ammanabrolu & Hausknecht; Yao et al. (2020); Atzeni et al. (2022) and use GRU-based *Vanilla TBRL* agent to evaluate our proposed approach (Figure 10 supplementary). We use individual GRUs Cho et al. (2014) for the information from the text-based environment such as observed text, the content of the inventory, the description of the room where the agent currently is located, and a valid list of actions. We learn the state representation by concatenating the individual representations from their GRUs Cho et al. (2014). We compute the action probability from both the

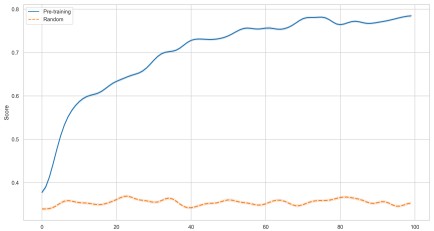 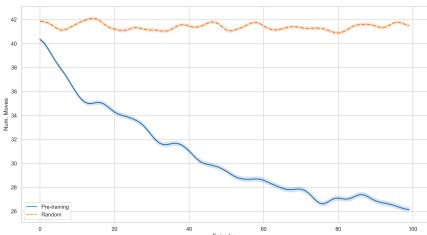

Figure 3: Training curves for pre-training step of STARLING depicting the normalized scores (left) and number of moves taken (right) of text-based reinforcement learning agents.

| Agents | Mean Normalized Score | Mean Moves Taken |
|---|---|---|
| Random | $0.050 \pm 0.01$ | $50 \pm 0.0$ |
| Pre-training | $0.72 \pm 0.063$ | $28.105 \pm 1.876$ |
| Human | $1.000 \pm 0.000$ | $9.640 \pm 5.620$ |

Table 1: Performance of random and pre-trained agents on a set of 25 unseen pre-training games after training on 75 pre-training games over 100 episodes. Mean Norm. Score (higher is better, normalized with maximum score achievable per game) and Mean Moves Taken (to achieve the goal, lower is better).

state and action representations. We use Advantage Action-critic (A2C) to train the network Mnih et al. (2016). All the results reported in this paper are averaged over 3 runs.

## 3.2 PRE-TRAINING TEXT-BASED RL AGENT

We generate 100 games that demonstrate basic skills in these environments such as *cooking pasta, painting the living room, boiling water, lighting a candle,* etc. We split these 100 games into 75 games for training and 25 for evaluation. We trained the vanilla TBRL agent on these 75 pre-training games over 100 episodes (50 max. steps per episode) and evaluated it on 25 held-out games. We compare the performance of the vanilla TBRL agent (pre-training) against both the random agent (picks random action at each step) and the Human performance. We use the mean normalized score and mean moves/steps taken by the agent for comparison. We collect the Human performance results based on the 48 participants (Section B). Figure 3 shows the training performance and Table 1 shows the evaluation results. We can see that human participants (high-school students) solved these games with a perfect normalized score of 1.0 indicating that these games are easy to solve. In order to successfully finish a game, an agent needs to take certain actions in a particular order. The order of actions taken decides the future states of the entities involved in the game.

## 3.3 SELF-SUPERVISED TRAINING OF TEXT-BASED RL AGENT

Next, we deploy the TBRL agent pre-trained on 75 games generated by LLM on different environments. We can the pre-trained TBRL agent as STARLING. We expect that STARLING will outperform the vanilla TBRL agent by utilizing the skills learned using LLM and boost the performance and generalization capabilities to reach the goal of the target environments: ScienceWorld, TWC and Zork1.

### 3.3.1 SCIENCEWORLD ENVIRONMENT

ScienceWorld environment evaluates the science reasoning abilities of the TBRL agents. It consists of several tasks from topics such as change of state, biological classification, etc. We select 4 tasks out of 30 tasks in the environment: task 1-1 changes of state (Boiling), task 4-1 finds a living thing, task 4-2 finds a non-living thing and task 5-2 grows a fruit. Each of these tasks contains $10 - 1400$ variations of the game and are split into $50\%$ training, $25\%$ for evaluation set, and $25\%$ for test set.

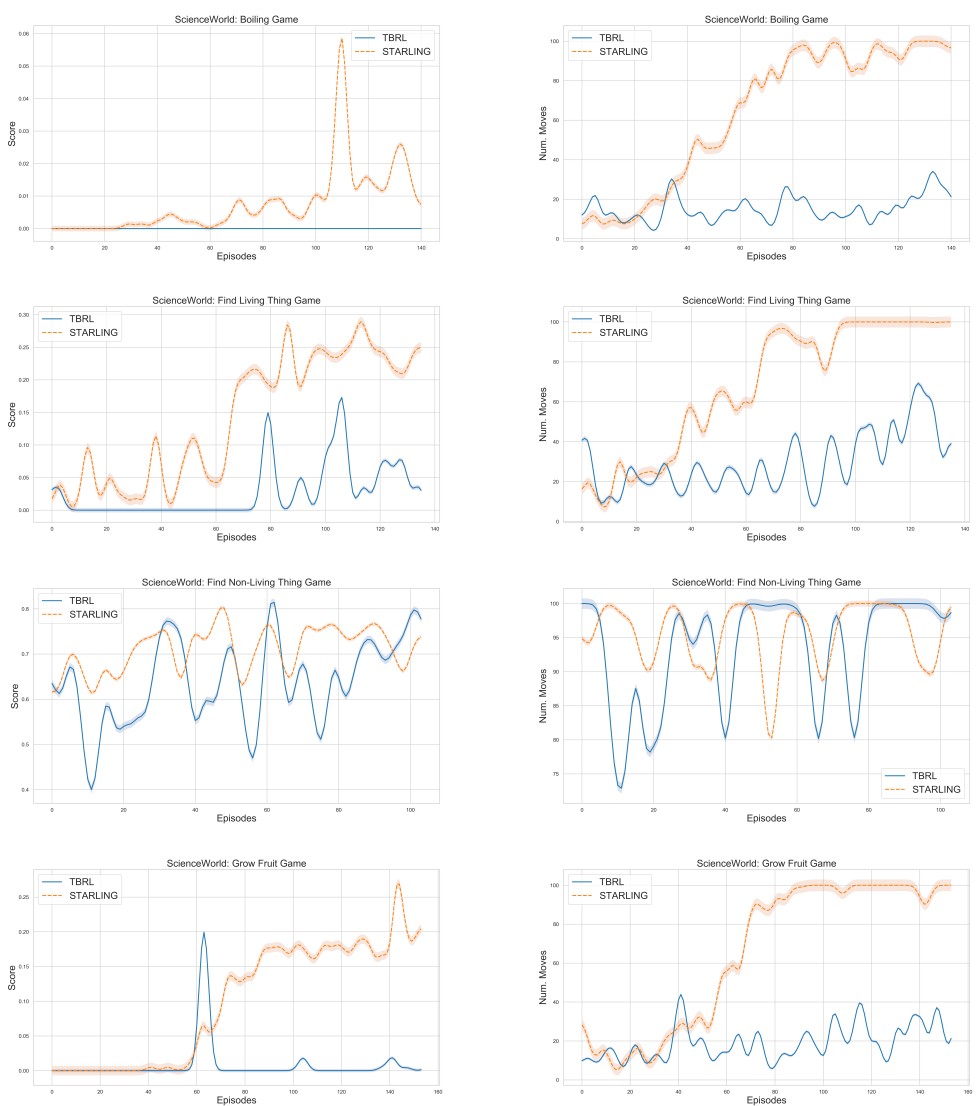

Figure 4: Training curves for ScienceWorld - Boil Substance (Task 1-1), Find a living thing (Task 4-1), Find a non-living thing (Task 4-2) and Grow a fruit (Task 5-2) games depicting the scores (left) and number of moves (right) of text-based reinforcement learning agents.

We train the STARLING agent with $100k$ maximum steps on a single environment (with a maximum of $100$ steps per game play)[3].

| Agents | ScienceWorld | | | |
|---|---|---|---|---|
| | Task 1-1 | Task 4-1 | Task 4-2 | Task 5-2 |
| *Vanilla TBRL* | $0.0 \pm 0.0$ | $0.25 \pm 0.0$ | $0.56 \pm 0.16$ | $\mathbf{0.21 \pm 0.02}$ |
| *STARLING* | $\mathbf{0.04 \pm 0.01}$ | $0.25 \pm 0.0$ | $\mathbf{0.94 \pm 0.06}$ | $0.09 \pm 0.05$ |

Table 2: Performance comparison of vanilla TBRL (without pre-training) and STARLING (pre-trained with 75 games generated using LLM) on test variations of $4$ tasks from ScienceWorld. All the scores and moves are averaged over 3 runs.

---

[3]Unlike in the "number of moves taken" metric in the pre-training results, the number of moves taken in the ScienceWorld measures how long the agent survived without reaching the failure state.

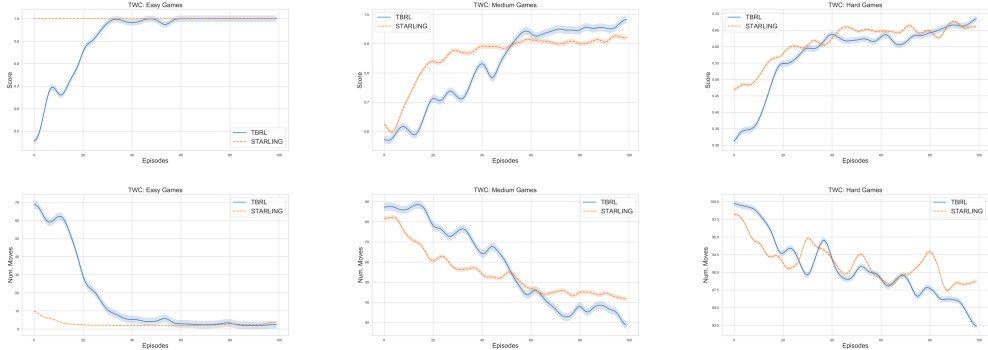

Figure 5: Training curves for TWC easy (left), medium (middle) and hard (right) games depicting the normalized scores (top) and number of moves (bottom) of both vanilla TBRL and STARLING agents.

| Agents | Textworld Commonsense | | |
|---|---|---|---|
| | Easy | Medium | Hard |
| *Vanilla TBRL* | $0.99 \pm 0.0$ | $0.81 \pm 0.06$ | $0.57 \pm 0.03$ |
| *STARLING* | **$1.0 \pm 0.0$** | **$0.85 \pm 0.04$** | **$0.64 \pm 0.05$** |

Table 3: Performance comparison of vanilla TBRL (without pre-training) and STARLING (pre-trained with 75 games generated using large language models) on the three difficulty levels of Textworld Commonsense (TWC). All the scores and moves are averaged over 3 runs.

Figure 4 shows the training curves for both the scores received and moves taken on the 4 tasks. We can see that STARLING outperforms vanilla TBRL on all the tasks. We notice that the pre-training steps improved the performance of STARLING in the first few episodes of tasks 4-1 (fina a living thing) and 4-2 (find a non-living thing). On the other hand, pre-training games such as boiling water, cooking pasta, planting a tree, etc. may have influenced the performance of STARLING in the later episodes of tasks 1-1 (change of state - boiling) and 5-2 (grow a fruit) by adapting the learned skills during pre-training to the target environment. we observe that STARLING learned to avoid failure state better than the vanilla TBRL.

Table 2 shows the performance comparison of both vanilla TBRL and STARLING on the test variations of the 4 tasks. We notice that we outperform vanilla TBRL on both task 1-1 and task 4-2. On the other hand, Vanilla TBRL does well in the task 5-2 (grow a fruit). We believe that the pre-training game on planting a tree lacked any specific knowledge about the role of pollinators when growing a fruit, to perform better on task 5-2.

### 3.3.2 TEXTWORLD COMMONSENSE ENVIRONMENT

TextWorld Commonsense environment evaluates the agent on commonsense reasoning about everyday objects such as toothbrush, dirty towel, etc. The environment, based on Textworld engine Côté et al. (2019), includes three difficulty levels: easy, medium, and hard depending on the number of objects to find and the number of rooms to explore. Each difficulty level includes 5 training games and 5 evaluation games similar to the distribution of the training games[4] for a total of 30 games with a batch size of 1 for this experiment. We train the STARLING agent on these 15 games for 100 episodes with a maximum of 50 steps.

Figure 5 shows the training curves of the three difficulty levels in the TWC environment. We can see that the STARLING agent gets a boost in performance both in the scores achieved and the moves taken compared to the vanilla TBRL. This shows that the pre-training step using LLM in STARLING leverages the basic skills mastered using the 75 generated games. Table 3 confirms our hypothesis

---

[4]In addition to the 5 evaluation games, TWC includes 5 test games from out-of-distribution. Since these games require external knowledge such as ConceptNet Speer et al. (2017), ATOMIC Hwang et al. (2021) etc., we exclude them from our experiments.

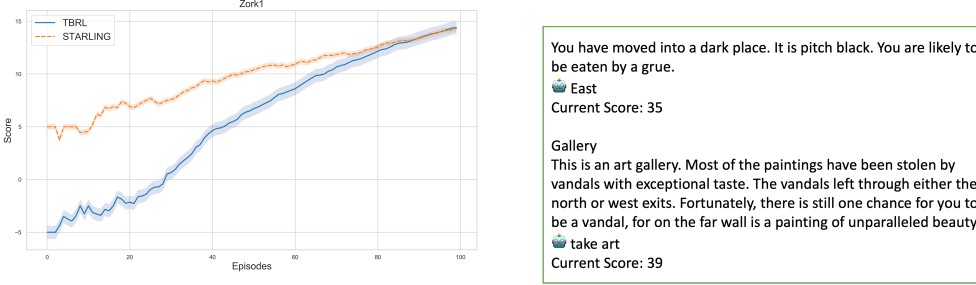

Figure 6: (left) Performance of TBRL agents on Zork1. (right) Sample trajectory from STARLING showing the bonus points of (+4 scores) for collecting the painting in the art gallery withing the first few episodes.

that the pre-training step in STARLING improves the overall performance across different difficulty levels.

### 3.3.3 ZORK1 ENVIRONMENT

Zork1 is a human-made interactive fictional game environment and one of the earliest known text-based games created based on the underworld characters with dark themes and characters such as dungeon, grue, elvish sword, etc. Zork1 is one of the 33 interactive games released as a part of the Jericho game suite. Unlike TWC and ScienceWorld, Zork1 includes a diverse set of locations (over 200 locations), larger action space, sparser rewards, and longer trajectories, making it a challenging environment.

In order to evaluate the effect of pre-training in the Zork1 environment, we simplified the game with "killing troll" as a final goal Zahavy et al. (2018). The agent needs to find the lantern and sword from the house, locate the hidden passageway to the underworld, light the lantern, and kill the troll. Without the lantern and sword, the agent entering the troll room reaches the failure state with negative rewards. In addition to these intermediate rewards, the game includes additional rewards when the agent collects a jewel-encrusted golden egg from the tall tree in the forest (+5 score) and a painting from the art gallery in the house (+4 score). We train the agents on 100 episodes with a maximum steps of 100 steps per episode.

Figure 6 (left) shows the performance of both the STARLING and vanilla TBRL on the Zork1 environment. As in TWC, we can see that the pre-training step boosts the performance of STARLING in the first few episodes compared to the vanilla TBRL agent. As in ScienceWorld, STARLING successfully avoids the failure state compared to the vanilla TBRL agent.

### 3.4 DISCUSSION

Our experiments on three text-based game environments show that pre-training these agents using LLM-generated games as auxiliary tasks generally boosts the performance of the agent. We notice that most of the performance gain achieved in the first few episodes of the gameplay on some game instances (e.g., training curves in ScienceWorld tasks 4-1 find a living thing and 4-2 find a non-living thing, TWC Easy and Hard, Zork1), whereas, STARLING adapts the basic skills learned during the pre-training to the target environment to improve the final scores (e.g., training curves in ScienceWorld tasks 1-1 change of state - boiling and 5-2 grow a fruit). We notice that STARLING avoids the failure states better than vanilla TBRL as can be seen in Zork1 and ScienceWorld. Similarly, STARLING tends to choose valid actions from the action space more effectively than vanilla TBRL (for example, see Figure 11 supplementary for the sample trajectories from ScienceWorld taken by vanilla TBRL and STARLING within the first few episodes for the task 1-1 changes of state - boiling).

Since these pre-training games lack navigational complexity that elicits skills such as planning, we observe that STARLING tends to suffer in games that require navigational skills (e.g., example, in ScienceWorld task 4-2 find a non-living thing, TWC hard difficulty and Zork1). Since the pre-training games involve fewer sequences of actions (short trajectories) to collect the reward and reach the final goal, STARLING struggles when the target environment has longer trajectories to reach the goal. On

the other hand, STARLING tends to collect bonus scores in Zork1 that are reachable within fewer steps instead of just chasing the larger rewarded states (e.g., see Figure 6 right).

# 4    RELATED WORK

## 4.1    TEXT-BASED GAMES

Text-based games provide a challenging benchmark for RL agents interacting with the environment in natural language. The common challenges for text-based RL are partial observability, combinatorial action space, sparse rewards, long-horizon planning, etc. To circumvent some of these challenges, it is typical that the games are often manually curated to evaluate a specific set of skills such as commonsense reasoning Murugesan et al. (2021a), knowledge graphs Ammanabrolu & Hausknecht; Murugesan et al. (2021b), exploration strategies Côté et al. (2019), etc. Environments such as TextWorld Commonsense Murugesan et al. (2021a) measure simple commonsense reasoning based on the one-hop relationship between a pair of everyday objects such (*apple* and *refrigerator*, *dirty towel* and *laundry basket*), but lacks diversity and complexity to learn a general set of skills. Environments such as ScienceWorld Wang et al. (2022) are often domain-specific environments that require domain knowledge to perform well in these environments. Jericho Hausknecht et al. (2020), on the other hand, includes human-generated games that require a complex set of skills to show any progress in the gameplay. These environments are manually created by humans with very limited automation in the variations of game generation by replacing similar or related objects, changing the layout/orientation of the environment, etc. Unlike these environments, STARLING provides an approach to leverage the skill generation capability of LLMs Huang et al. (2022) to automatically generate text-based games based on the input game ideas with minimal human supervision. These games are generated automatically by requesting a specific set of game-related facts from LLM in a slot-filling style text generation Rakotonirina et al. (2022).

## 4.2    SELF-SUPERVISED RL

Self-supervised RL has been a popular topic in vision-based RL and robotic environments Sekar et al. (2020); Li et al. (2022). To the best of our knowledge, we are the first to utilize LLM to generate the games to train text-based RL agents[5]. Previous works have utilized LLM for action generation Yao et al. (2020), play interactive fictional game Tsai et al. (2023), build a world model Ammanabrolu et al. (2020), etc. These works showed that using LLM to learn the underlying representation of text in the environment does not necessarily improve the performance (Wang et al., 2022). It is shown that novel exploration strategies and efficient RL algorithms along with the learning model similar to the one in Figure 10 outperform all the other LLM-based agents Tuyls et al. (2021). On the other hand, generalist agent has been recently explored to generalize across multiple environments Reed et al. (2022), but the performance of these agents on a diverse set of environments are less convincing Cobbe et al. (2019).

# 5    CONCLUSION

In this paper, we proposed a novel self-supervised training for a text-based reinforcement learning agent, STARLING, with the help of the generalized skill generation capability of large language models like GPT3. We generated a set of text-based games that require agents to learn basic skills such as cooking pasta, boiling water, etc., and utilize sequential decision-making over the modality of text. The proposed STARLING uses the GPT-3 pre-trained language model to automatically generate these games. This approach can be used to create additional games or adapted to build games for new domains with minimal human intervention. We showed that the STARLING agent pre-trained on the games generated by LLM outperforms vanilla TBRL. We evaluated STARLING on three environments: ScienceWorld, TWC, and Zork1. In all these environments, STARLING showed enhanced skills in the target environment.

---

[5]Several LLM-based interactive fictional environment for entertainment purpose exists AID (2019).

## 6 LIMITATIONS

Human participants were volunteers from a local High School that agreed to participate in this study. This may have introduced a bias into the human participant data since all participants were high school educated, from one geographic region, between the ages of 15 and 18, and volunteers. Many of these participants complete homework assignments and assessments often which may make their reasoning skills better than potential participants outside of school. In the future, testing human participants from various geographic locations, age groups, and levels of education may reduce bias. The STARLING currently requires human intervention and/or the Game Validator (from the glulx compiler) to build functioning games. We will continue to work on building an end-to-end version of the STARLING, that can take a game idea and turn it into an interactive fiction game without any human intervention. This would speed up development time so a larger set of games can be created.

Large language models such as ChatGPT have been developed recently with the ability to interact with users in a manner conversationally similar to the interactions found in interactive fiction games. From our experimentation, ChatGPT struggles to keep track of the states of all in-game objects and the pre-conditions necessary to use those actions (e.g. ChatGPT does not always require the player to turn on the stove before using it) as well as Inform7-based games. In addition, it suffers from small factual errors, and is hard to reproduce the same result, through this could also be seen as a benefit. Despite these challenges exploring the use of models such as ChatGPT to interact with agents shows promise in the future.

## 7 ETHICS STATEMENT

We asked the human participants to play the games generated by STARLING to evaluate the game's complexity and clarity. After receiving IRB approval from a local High School and informed consent from each of the 48 human participants, we asked the participants to play five randomly assigned games via `iplayif.com`, an online interactive fiction game player. Players received the goal of the game and the list of admissible actions. We collected the number of steps that each player took and the score received for each game via Google form. We did not collect any personal information or personally identifiable information as a part of this study.

Since we use large language models such as GPT3 to generate a set of text-based games, the bias and other fairness/ethical concerns that come with the LLM may unintentionally transfer to the pre-trained agent. Additional mitigation steps may be required to filter harmful contents from the generated response.

## 8 REPRODUCIBILITY STATEMENT

As an effort to encourage further research in self-supervised text-based RL, we plan to release the 100 games generated as a part of this paper, the source code for STARLING, script to generate game-related files based on a set of game ideas and LLM (including game templates and metadata) as an open-source project.

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

## A  ADDITIONAL DETAILS

In order to generate games that require composing previously learned skills, we take inspiration from household chores, cooking, and maintenance tasks. We generate 100 game ideas and use the STARLING to generate a set of 100 games. We choose the game ideas carefully for the learning agent to utilize similar skills (ex. baking, mixing, spreading, using a hammer, etc) in new situations therefore forcing the agent to generalize skills and compose them with other skills. For example,

while cooking pasta, an agent must learn how to boil water which is a skill that can be applied for a related game idea "brewing tea".

LLMs such as GPT-3 are prone to making factual and grammatical errors, in addition to violating the specified format. We check for any errors in the generated game(s) using a Game Validator as a part of *glulx* compiler which uses depth first search (DFS) to explore all the possible trajectories in the game. To correct for minor errors and inconsistencies in each game, information from GPT-3 can also be optionally verified by the human authors in the JSON file. We found that, in cases when the created game has errors, restarting the game generation a few times usually results in a playable game.

## A.1 FROM GPT3 OUTPUT TO GAME JSON

We extract the information from GPT-3 using Python simple regular expression rules by first splitting the output into three sections: task sequence (ex. Open cabinet, take pot), objects (ex. Pot), and actions (ex. Fill pot with water). We add the task sequence to the list of admissible actions the player could execute within the game. We store the objects internally with a type, a location, a name, and a set of properties. We further split the actions section into default actions and custom actions which are actions native and non-native to inform7 respectively. Similar to the objects, we store each custom action internally with a name, a declaration, a definition, a set of constraints, a set of prerequisites, and a set of post-requisites.

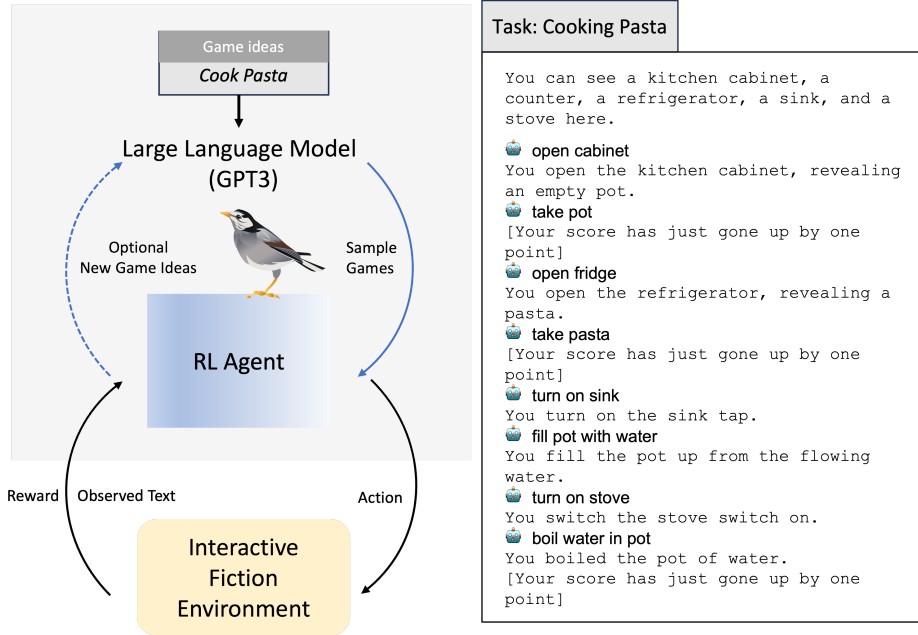

Figure 7: (Left) shows the overview of the proposed self-supervised text-based reinforcement learning with large language model. (Right) shows the sample text-only agent play through of cooking pasta game. Players must use the boil skill at the correct time to be successful.

## A.2 MODIFYING TEXTWORLD GYM FOR STARLING

OpenAI Gym is a general reinforcement learning framework that acts as an interface between RL agents and Inform7-based STARLING game engine Brockman et al. (2016). Gym connects environments with agents by using a monitor to keep track of every step, state of the game, the final score of agents, and the sample complexity or the amount of time an agent takes to learn. Most default environments in Gym support a continuous or discrete action space although interactive fiction games require combinatorial action spaces in natural language Hausknecht et al. (2020). The TextWorld Gym customized the OpenAI Gym for interactive fiction games. In this work, we repurpose the custom Gym environment created for TextWorld environment with Inform7 object and action types.

TextWorld's Gym environment only supports TextWorld-generated games which includes a Glulx compiled game file and a TextWorld-generated JSON file with game metadata defined in proprietary TextWorld classes. This restricted our ability to create games with objects and actions previously undefined in TextWorld environment. These objects and actions must be defined according to TextWorld's grammar and logic rules. This is a time consuming process and is prone to many errors. The goal of the STARLING Game Generator is to allow users to automate the game creation using LLM, and most importantly, create games without learning a new programming language or familiarizing themselves with any grammar rules.

To get rid of these restrictions, an entirely new wrapper was created which acted as an interface between the game engine and TextWorld Gym environment. This wrapper ensures that the user to freely define any object or action type and the environment works with any Glulx compiled game file without dependence on the TextWorld-generated metadata to track the state of objects throughout the game. The wrapper does this by parsing the observation state returned by game engine after every step to generate certain data-points like admissible commands, current score, last action, number of steps taken and inventory required by the TextWorld Gym environment.

## B  HUMAN PARTICIPANTS

Humans are considered to have exemplary compositional skill learning so comparing their performance to pre-trained agent's performance is valuable to validate generated games's difficulty and effectiveness as a pre-training task. After receiving IRB approval from a local High School and informed consent from each of the 48 human participants, we asked the participants to play five randomly assigned games via `iplayif.com`, an online interactive fiction game player. Players received the goal of the game and the list of admissible actions. We collected the number of steps that each player took and the score received for each game via Google form. [6]

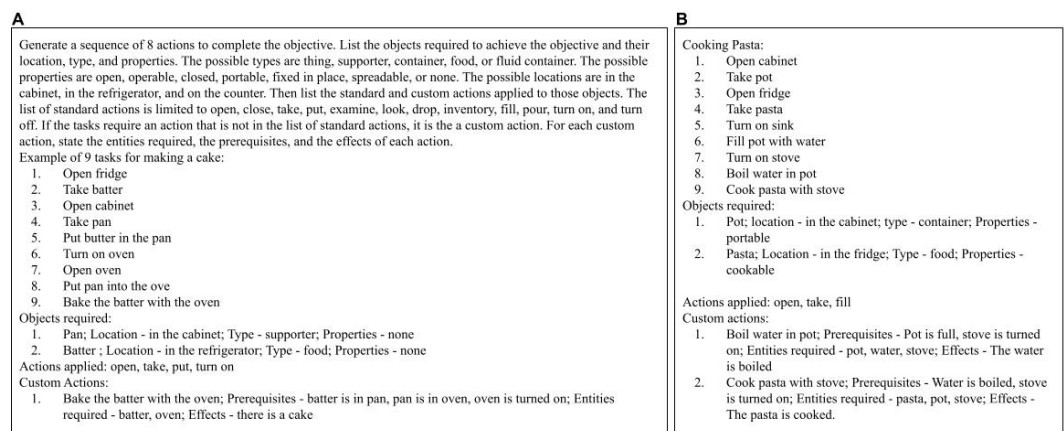

Figure 8: (A) GPT-3 input prompt for cooking games with one action example. The actual prompt contains four action examples. (B) GPT-3 output for cooking pasta game idea. GPT-3 reliably outputs accurate and necessary game information very similar to the input.

---

[6]No personal information was collected as a part of this study.

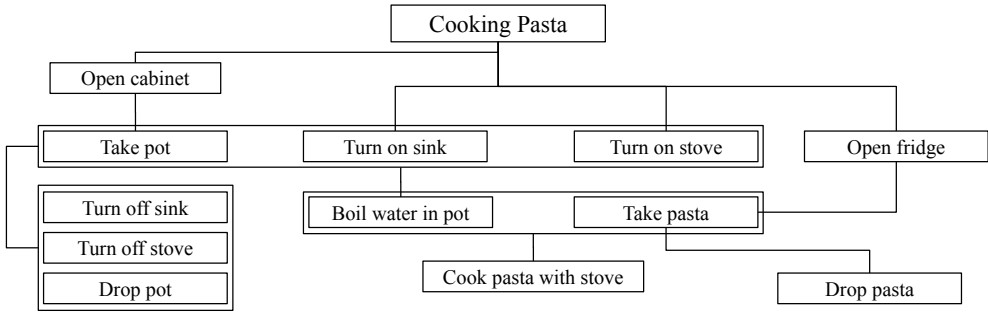

Figure 9: Composition of skills needed to complete the game "*Cooking Pasta*" as Flow diagram. A line between two skills represent that one skill needs to be executed before executing the other one (E.g., *Open cabinet* ⟶ *Take pot*). A box with multiple skills represent that skills within the box can be executed in any order (E.g., *Boil water in pot* ∥ *Take pasta*).

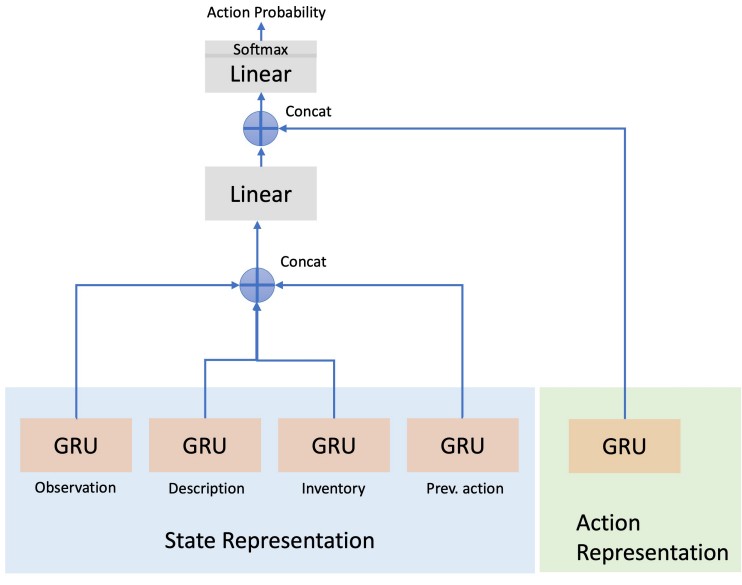

Figure 10: Vanilla Text-based RL agent used in this paper.

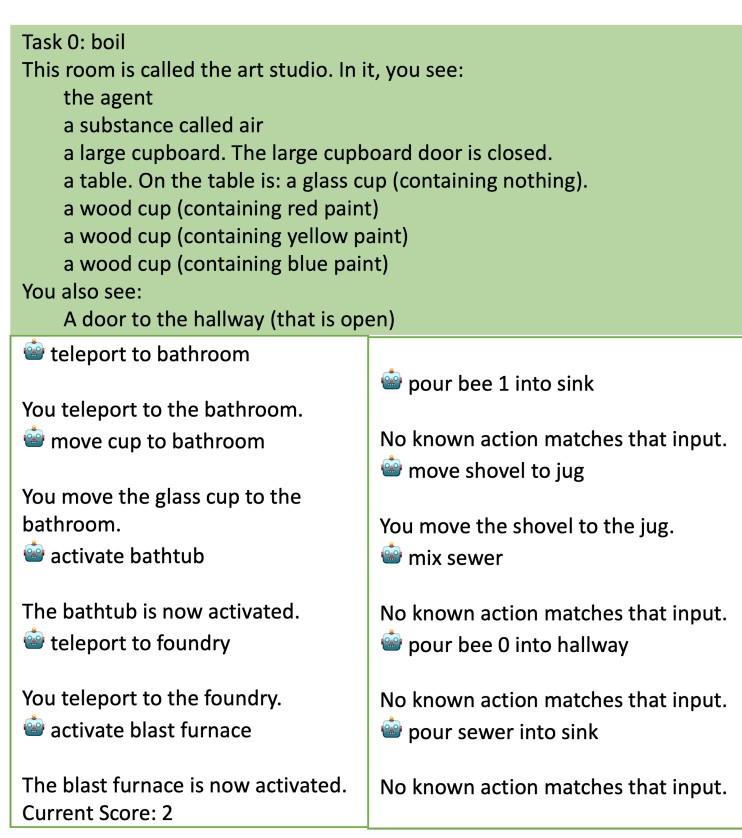

Figure 11: Sample Trajectories taken by STARLING and Vanilla TBRL for the task 1-1: change of state (boiling) task in ScienceWorld.

```
{"libraries" : [
    {"name": "measured liquid",
     "author": "Emily Short"},
    { "name": "modern conveniences",
     "author": "Emily Short"}],
"modules" : ["scoring"],
"room" : { "name": "home kitchen",
    "description": ""},
"custom entities" : [ "Food is a kind of thing. Food is usually edible. Food can be raw or cooked. Food is usually raw."],
"entities" : [{"name": "pot",
     "type": "container",
     "properties": ["portable", "open"],
     "location": "in the cabinet"},
    { "name": "pasta",
     "type": "food",
     "properties": "",
     "location": "in the refrigerator"},
    {"name": "sauce",
     "type": "food",
     "properties": "",
     "location": "in the refrigerator"}],
"scoring" : [
    {"condition": "taking the pot for the first time",
     "increment": "1"},
    {"condition": "taking the pasta for the first time",
     "increment": "1"},
    {"condition": "taking the sauce for the first time",
     "increment": "1"}],
"actions" : [{"name": "",
     "declaration": { "command": "cook [something] with [something]",
       "alias": "cooking it with",
       "applicable_to": "one carried thing and one thing"},
     "prerequisites": [],
     "constraints": [{"condition": "the noun is not a food",
         "prompt": "You can't cook that."},
       { "condition": "the second noun is not a stove",
         "prompt": "You can't cook that."}],
     "definition": { "tasks": [ "increase score by 1"],
       "prompt": "You cooked the [noun] with the [second noun]." },
     "postrequisties": [] } ],
"end_game" : {
    "condition": "4",
    "task": "end the story finally",
    "tasks": ["look", "inventory", "open cabinet", "take pot", "drop pot", "open fridge", "take pasta", "drop pasta", "turn on
sink", "turn off sink", "fill pot with water", "turn on stove", "turn off stove", "boil water in pot", "cook pasta with stove"]}}
```

Figure 12: Example JSON file produced for cooking pasta game idea. The libraries, modules, and room sections were part of the setup, the custom entities and entities sections correspond to object creation, the actions correspond to the custom actions, and the scoring and end game correspond to the rewards sections of each game. The entities section describes names, types, and properties of entities present in the game. The actions section defines custom actions including their declaration, alias, and constraints not part of Inform7 by default. The end-game section defines the maximum score and the list of admissible actions that the user can take.

