# OpenReview forum: "STARLING: Self-supervised Training of Text-based Reinforcement Learning Agent with Large Language Models"
_ICLR.cc/2024/Conference — ICLR 2024 Conference Withdrawn Submission_

### Official Review · Reviewer_vFed · 2023-10-26

**Soundness:** 2 fair
**Presentation:** 3 good
**Contribution:** 2 fair
**Rating:** 3
**Confidence:** 4

**Summary:**

The paper focuses on improving the generalization capabilities of text-based reinforcement learning (TBRL) agents through an interactive environment called STARLING. This environment uses large language models (LLMs) like GPT-3 and an interactive fiction game engine (Inform7) to automatically generate text-based pre-trained games. These games help TBRL agents to hone specific skills and improve their performance in various tasks.

**Strengths:**

1. The paper is well-structured and easy to follow.

**Weaknesses:**

1. The paper's focus on using LLMs to generate configuration files for a specific game engine may not offer broad insights applicable to other related work.

2. The paper's use of text-based pre-trained games may not generalize well to other forms of reinforcement learning (RL) tasks. Moreover, high-performance LLMs could potentially solve text-based tasks directly, questioning the need for pre-training to acquire skills.

3. The lack of comparative experiments with other pre-training methods makes it difficult to ascertain the advantages of using LLMs for designing pre-trained games.

**Questions:**

I have no further questions.

---

> ### Author Response · Authors · 2023-11-21
> **We thank the reviewer for going through the paper in such detail and giving us their valuable feedback. We have tried to address all the concerns raised by the reviewer in the following response.**
>
> 1. Even though we are using the LLMs to create configuration files, each file contains the objects, actions, and skills that are necessary to certain games. This is like the approach from ScienceWorld which uses configuration files to specify game details.
> 2. Though we could use LLM’s here, they require lots of compute power than a simpler model. This is especially tune if we want to fine tune the models to a specific domain. It is much less compute-intensive and cheaper than using an LLM to solve the text-based tasks directly.
> 3. Can the reviewer provide alternative pre-training methods. The pre-training method that we used models how the agent would act in a test scenario. If we want the agent to learn specific skills, the agent plays our LLM-based games to learn these skills which can then be applied later in future environments.

---

### Official Review · Reviewer_2y7E · 2023-10-27

**Soundness:** 3 good
**Presentation:** 2 fair
**Contribution:** 2 fair
**Rating:** 5
**Confidence:** 5

**Summary:**

In this work, the authors propose an LLM-aided method that generates text-based game environments, that are similar to some target domain text-based games, to pre-train RL agents. The authors show that compared to agents trained on target games from scratch, RL agents pre-trained on the generated games could benefit from having a better initialization, either learn to solve games more efficient, or to achieve more scores.

The authors use three different text-based game suites, namely ScienceWorld, TextWorld Commonsense, and Zork 1. In which, ScienceWorld and TWC consists of synthetically generated games, Zork 1 is human-authored game that is significantly more difficult.

**Strengths:**

* Interesting topic: I find the overall idea of leveraging large foundation models (e.g., GPT-x) in helping downstream tasks (especially interactive tasks) thrilling. Text-based games are interactive environments where state information can be represented in pure text, and thus they can be a good testbed to study LLM applications in an interactive setting.

* Interesting approach: There are some other works in the field that aim to use LLMs 1) as an actor (i.e., given a piece of text describing the state, one queries an LLM to suggest the next action); or 2) as a critic (ask an LLM to review a trajectory in a hindsight, and use the resulting feedback or score as additional training signal to improve the agent). Different from existing work, here, the authors propose to leverage LLMs in a data augmentation manner. In such way, we can harvest knowledge from the LLMs and represent these knowledge as interactable environments (e.g., how to cook pasta). This is where I'm the most excited because in comparison with extracting static knowledge from LLMs (e.g., `cooking pasta requires a pan`), this approach generates environments that an agent could really interact with, through exploration, trial an error, the agent could learn such knowledge in an arguably more grounded way. In this work, the downstream task is to solve text-based games, it is also natural to augment the data in the form of games.

* Human score: I appreciate the authors evaluated the generated games with human annotators.

**Weaknesses:**

Please see below in the questions section.

**Questions:**

### Concerns and Questions
1. The current version of the paper (mainly Section 2) lacks the clarity on how exactly the games are being generated. From my understanding, not all steps described in Figure 2 (left) is performed automatically. For instance, how exactly the authors turn JSON files into Inform7 code, is that done by the LLM or the authors manually? On a related note, the authors mention `We compile this code using the Glulx...` but to my understanding Glulx is just an interpreter, not a compiler. Could the authors clarify this?
2. I like the authors decompose text-based games as trees of `skills`. To make the statistics of generated games more clear, I'd suggest the authors to provide more information on the diversity of the skills, and how these skills relate to skills appear in ScienceWorld/TWC/Zork? The authors hypothesize that RL agents may have adapted skills learned from pre-training games to target games, but there is insufficient evidence on why/how this is the case.
3. Related to the previous point, how exactly do the authors obtain `game ideas`, or the `seeds`? This could be quite important because the seeds may have big impact on the distribution shift between generated games and the target games. I.e., if the distributions are close enough, it makes sense that the augmented games could be helpful. But otherwise, there might be other reasons why they are helpful.
4. Is there a specific reason why the variance in the figures (e.g., Figure 4) are so low? It seems all the 3 agents with different seeds learn in a very similar way?
5. Figure 6 is a bit confusing, I don't understand why in the example on the right the agent gets 35 and 39, but in the left figure the scores at the end is 15.
6. In Figure 5 (mid and right), it seems the vanilla baseline end up getting higher training scores, but in Table 3, STARLING achieved better evaluation scores. Is that because vanilla baseline is overfitting? The author say in Section 3.3.2 that `We can see that the STARLING agent gets a boost in performance both in the scores achieved and the moves taken compared to the vanilla TBRL.` But this is true only at the beginning of the training.
7. In addition to my concern that the current version of the submission has insufficient clarity on game generation and insufficient analysis on the generated games, I feel the provided results are not strong enough. The vanilla TBRL seems to be a weak baseline, but even that, STARLING is not outperforming TBRL by a large margin. For instance, on Zork1, given enough episodes, vanilla TBRL could reach the same score as STARLING, given the fact that STARLING requires an extra 100 episodes of pre-training.
8. I have a mixed feeling of raising this point, but in some recent works such as [SwiftSage](https://arxiv.org/abs/2305.17390), agents could achieve 0.97 test score on ScienceWorld task 1-1, compared to 0.04 here. I fully understand that these are totally different approach (maybe even orthogonal) and is less appropriate to be compared directly, but I encourage the authors to think how the data augmentation approach could be used to facilitate modern and much stronger baseline systems (than an RNN-based RL agent). Related to my previous concern, I am a bit hesitant when I see results of task 1-1 in Table 2, comparing 0.04 with 0.

### Typos and minor things
1. There are two TextWorld bib entries (page 11). The correct one is Cote et al.
2. At the bottom of page 3, there are two sentences with repeating info: ` If the user approves the game metadata in the JSON file, we write and compile the Inform7 code based on the JSON file into an Inform7 game for a given game idea. If the user approves the game-related data in the JSON file, we write and compile the Inform7 code based on the JSON file into an Inform7 game for a given game idea. `
3. In Section 3.3: `We can the pre-trained TBRL ...`
4. In page 7, under Table 3: `... (fina a living thing ) ...`
5. Figure 6 caption: `... in the art gallery withing the first few episodes.`
6. Please use a larger font size in all figures.

### Suggested citations
1. [ByteSized32: A Corpus and Challenge Task for Generating Task-Specific World Models Expressed as Text Games](https://arxiv.org/abs/2305.14879). In this work, Wang et al. also explore to generate text-based games with the help from LLMs. The approach and goal are not exactly the same as in here, but still worths discussing at least in the related work section.

---

> ### Author Response · Authors · 2023-11-20
> **We thank the reviewer for going through the paper in such detail and giving us their valuable feedback. We have tried to address all the concerns raised by the reviewer in the following response.**
>
> 1.	We have created a rule based engine that converts the JSON files to inform7 code. The objects in the JSON generated by the LLM are configured to reflect all important aspects required to generate the inform 7 code for a game. The rule-based engine uses these objects to generate the required lines of code.
> Yes, Glulx is an interpreter that we use to run the generated inform 7 code after it’s been compiled by the inform 7 engine. We thank the reviewer for pointing out this mistake. We will rectify this mistake in the updated draft of the paper.
>
> 2. The pre-training games included various categories of skills to be picked up by the agent in order to complete the games. These tasks included skills like cooking, mixing two substances, planting a tree, finding hidden objects, boiling a liquid, etc. A similar set of skillsets (but, in a different setting) were required in the chosen games like finding a living/non-living thing, growing a fruit, etc. Figures 4 and 5 along with tables 2 and 3 compare the performance of a pre-trained STARLING agent with that of a Vanilla TBRL. The comparison shows that the pre-trained agent outperforms the Vanilla one. Thus, showcasing that the pre-trained agent is able to pick-up on the required skills for completing the games.
>
> 3. The game ideas are picked up based on certain skills that we would like the agent to pick up. The idea here is to create an environment that could be used to generate a set of games to teach the agent a certain category of skill. For e.g we used the game generation module to create games that require the agent to perform a sequence of complex tasks to achieve the goal like following a recipe for cooking, finding required tools for planting a tree, etc. The objective here is to showcase that the environment can be extended or used to leverage the generative capability of LLMs to teach any skill to the RL agent.
>
> 4. The variance in the figures are low because of the difficulty or complexity of the games. Even after the pre-training step, although the performance of these agents improve, but they still aren’t able to perform well on all the tasks except for task (4-2).
>
> 5. We thank the reviewer for pointing this out. The example on the right was meant to just showcase the kind of steps/actions taken by the agent. It is from a similar experiment but, does not belong to the exact experiment show-cased on the left.
>
> 6. Yes, the reason for this is due to the over-fitting of the vanilla baseline. It has poor generalisation capabilities. The TWC games are relatively simpler and easier than the ScienceWorld games. This is the reason for the stark difference in performances of the two agents on ScienceWorld games when compared to TWC games.
>
> 7. The main objective here is to showcase the generalisation capabilities of the STARLING baseline. Also, the ability of the game generation module to generate games for fostering development of agents that can acquire skills which can be leverage across environments. The same pre-trained STARLING agent was able to perform better across Zork, Science-World and TWC. We agree with the reviewer that perhaps a stronger baseline than TBRL to increase the difference even further.
>
> 8. We thank the reviewer for this point. Yes, as discussed in the point above, perhaps a stronger baseline can be used to increase the difference between the pre-trained and Vanilla agents. The can help make a stronger case for the main objective too, which is to showcase the generalisation capabilities of the STARLING baseline/
>
> We thanks the reviewer for pointing out the typos and minor things. We will address all of them in the updated draft.

---

> > ### Comment · Reviewer_2y7E · 2023-11-20
> >
> > Thank you for addressing and clarifying my questions. I look forward to seeing the next versions of this paper.
> >
> > At this point, changing my score would less likely to make any difference, unless some active reviewer-author discussion happen within the next 1 or 2 days. So let me just drop more thoughts here.
> >
> > In addition to integrating the proposed method with stronger baseline systems/agents, another potential direction is to generate text-based games on the fly while an agent interacting with a 3d environment. In such sense, the generated text environment can serve as a discrete and abstract world model, and importantly, it is interactable. Similar to the motivation from many text-based game research, here we can assume that 1) many reasoning/planning can be learned and practiced in an abstract level (i.e., language) then applied to 3d world; and 2) the reasoner plays/learns in this language environment could easily leverage knowledge embedded in pre-trained LLMs (hopefully this is still true when gpt-4v is available).
> >
> > Overall I see quite some potentials in the direction the authors pursue.

---

### Official Review · Reviewer_yam5 · 2023-10-31

**Soundness:** 2 fair
**Presentation:** 3 good
**Contribution:** 2 fair
**Rating:** 3
**Confidence:** 4

**Summary:**

In this work, the authors propose STARLING, an interactive text-based environment that leverages large language models (LLMs), such as GPT-3, to automatically generate text-based games. These games are created based on a seed set of game ideas and are designed to help RL agents acquire specific skills. The generated games involve tasks like cooking pasta, boiling water, and more, requiring RL agents to perform a sequence of actions to achieve a goal.

They evaluated STARLING on three environments: ScienceWorld, TWC, and Zork1. In all these environments, STARLING showed
enhanced skills in the target environment.

**Strengths:**

1. It presents an interesting pipeline to leverage LLM to pre-train RL. They prove that STARLING outperforms the baseline RL agent by utilizing the skills learned using LLM and boosting the performance and generalization capabilities to reach the goal of the target environments: ScienceWorld, TWC, and Zork1.

2. They promise to release the 100 games generated as a part of this paper, and the scripts to generate game-related files based on a set of game ideas and LLM. I think these prompts will be helpful to RL agent's training and test.

**Weaknesses:**

1. Concern about overlapping between training and test datasets. B/c LLMs were trained on many web texts which include text adventure game transcripts. When you test your pre-trained model on some existing games, will this cause data leaking?

2. Limited Comparison. Only compare the model with a vanilla text-based RL agent. You use LLM, so whether an LLM-based agent will be worth comparing to?

3. Scalability and Complexity: Compared to the vanilla text-based RL agents, what is the  Scalability and Complexity of STARLING?

4. Experiments: I only found some limited comparisons under 100 episodes. I did not see the results after converging.

**Questions:**

See weakness.

1. how to make sure the LLM games/skills you use for pre-training has no overlapping or similar scenario of test games?

2. Compared to an LLM-based agent, what will be the results?

---

> ### Author Response · Authors · 2023-11-21
> **We thank the reviewer for going through the paper in such detail and giving us their valuable feedback. We have tried to address all the concerns raised by the reviewer in the following response.**
>
> 1. The environments are designed in such a manor where the agent must pick from a pre-determined list of admissible actions rather than like a QandA style response. Therefore, data leaking is not a concern here.
> 2. LLM’s have been trained on a wide set of internet data already and likely already have generalization capabilities. Therefore, an LLM agent likely would not show the generalizability. By using a simpler and less complex model we show how agents can learn generalizability while being less complex. LLM’s are costly and large models, we are demonstrating generalizability among smaller, cheaper, and less complex models.
> 3. We picked only 100 episodes because the TBRL agents started to overfit after 100 episodes. The games are complex but not complex enough to warrant training past 100 episodes without the risk of overfitting.
> 4. We ensure there is no overlapping in the games that are generated. We want the agent to learn skills from the environments. The games that we create have different skill sets that are required. Since each game requires a different skill set, we know there is no overlap between the games.

---

### Official Review · Reviewer_Fptt · 2023-11-01

**Soundness:** 2 fair
**Presentation:** 1 poor
**Contribution:** 1 poor
**Rating:** 3
**Confidence:** 4

**Summary:**

This paper introduces STARLING, a self-supervised training of text-based reinforcement learning agent with large language models. Specifically, the authors propose to utilize a LLM (GPT3) to and an interactive fiction game engine (Inform7) to generate text games based on some seed ideas. This is motivated by the current limitations of text-based games for RLM studies. including simple games lacking complexity, and being domain-specific without generalizable capability. The generated games are used to pre-tran text-based RLM agents before evaluating on downstream tasks including ScienceWorld, TWC, and Zork1. Results show enhanced skills in the target environments.

**Strengths:**

1. This paper introduces a method to augment pre-training data for text-based games. The data may mitigate previous limitations such as low complexity and domain-specific.
2. The augmented data may improve model performance on downstream tasks.

**Weaknesses:**

1. There are many details and analysis missing. Please see the questions section below.
2. Some key claims are not justified. The paper is motivated to generate data with high game complexity and generalized skills to other situations. However, from the evaluation and discussion, STARLING "lack navigational complexity that elicits skills such as planning", and "involves fewer sequences of actions" (although specifically designed to have a minimum number of actions). These cannot show the benefits of the pre-training.

**Questions:**

1. In Section 3.3., what skills are learned "using LLM"? My understanding is that you only use the LLM to generate games, while the agents are trained from scratch (according to Section 3.1)
2. why did you select these 4 tasks out of 30 tasks from SCIENCEWORLD?
3. Did you consider the 25 evaluation games as the development set?
4. How did humans perform the study? Were they instructed to complete the tasks directly?
5. Why would pre-training games may have influenced the performance of STARLING in the later episodes of tasks 1-1 and 5-2"? Why would the performance comparison between the vanilla TBRL and STARLING vary dramatically among the four tasks? If "STARLING" really learned to avoid failure states better, why would the performance drop? And can you clarify what you mean by "avoid failure states"?
6. For ZORK1, does the result suggest that pre-training helps with initialization, but in the end perform similarly to the baseline? What are the results comparisons in the test set?


Suggestion:
1. Please use \citet to cite references
2. Please double check your writing. There are many typos and errors that make it really hard to follow the paper. For example, there are sentence repetitions at the bottom of page 3.

---

> ### Author Response · Authors · 2023-11-21
> **We thank the reviewer for going through the paper in such detail and giving us their valuable feedback. We have tried to address all the concerns raised by the reviewer in the following response.**
>
> 1. The skills learned using LLM refers to the skills learned from playing in the environments created by the LLM. This is an error in the wording of the paper which we will rectify.
> 2. We picked these 4 tasks because they represented complex and varied skills that would be necessary to be successful. These 4 skillsets can be extended to others and require agents to be generalizable. This is shown by the generalizability of the agents when they use pre-training.
> The pre-training games included various categories of skills to be picked up by the agent in order to complete the games. These tasks included skills like cooking, mixing two substances, planting a tree, finding hidden objects, boiling a liquid, etc. A similar set of skillsets (but, in a different setting) were required in the chosen games like finding a living/non-living thing, growing a fruit, etc. Figures 4 and 5 along with tables 2 and 3 compare the performance of a pre-trained STARLING agent with that of a Vanilla TBRL. The comparison shows that the pre-trained agent outperforms the Vanilla one. Thus, showcasing that the pre-trained agent is able to pick-up on the required skills for completing the games.
> 3. The 25 games were the test set. The 100 games were generated from the same distribution and were randomly split 75-25. 75 was the training set and 25 was the test set. What is meant exactly by the development set?
> 4. Humans were given five randomly selected environments from the set of 100 generated environments. They then played through those environments based on the actions that made sense. The environments were presented as Text-based games such as Zork. The human participants chose actions to take in the environment from a list of admissible actions until they reached a failure state or successfully completed the objective of the environment. The number of actions or steps and score that each human received was recorded.
> 5. The nature of ScienceWorld games includes dead-states/failure-states which is a wrong action that the agent can take that is irreversible. If the agent takes these steps they cannot complete the objective of the environment. An example of this is burning a necessary food ingredient to a certain recipe. The agent cannot reverse this action and leads to a failure state.
> The performance of these games varies dramatically due to how different and complex the distribution of games are. Even after the pre-training the agents is given the option of many options to take. of steps to complete further indicating how complex the games are. The performance drops at a later stage
> Since STARLING has been pre-trained, it has better generalization abilities as it is able to better avoid failure states rather than completely avoid them.
> 6. Yes, pre-training helps with initialization not just for Zork, but for TWC and other environments as well. Even though the vanilla performs similarly to STARLING in the training set, STARLING performs better in the test set indicating its superior generalizability.

---

> > ### Comment · Reviewer_Fptt · 2023-11-22
> >
> > Thanks for the responses.
> >
> > I still believe that more revision is needed (including the details I mentioned in addition to the author response such as how LLMs are learning the skills) to justify the claims.